# A GENERALIZED EIGENGAME WITH EXTENSIONS TO DEEP MULTIVIEW REPRESENTATION LEARNING

## ABSTRACT

Generalized Eigenvalue Problems (GEPs) encompass a range of interesting dimensionality reduction methods. Development of efficient stochastic approaches to these problems would allow them to scale to larger datasets. Canonical Correlation Analysis (CCA) is one example of a GEP for dimensionality reduction which has found extensive use in problems with two or more views of the data. Deep learning extensions of CCA require large mini-batch sizes, and therefore large memory consumption, in the stochastic setting to achieve good performance and this has limited its application in practice. Inspired by the Generalized Hebbian Algorithm, we develop an approach to solving stochastic GEPs in which all constraints are softly enforced by Lagrange multipliers. Then by considering the integral of this Lagrangian function, its pseudo-utility, and inspired by recent formulations of Principal Components Analysis and GEPs as games with differentiable utilities, we develop a game-theory inspired approach to solving GEPs. We show that our approaches share much of the theoretical grounding of the previous Hebbian and game theoretic approaches for the linear case but our method permits extension to general function approximators like neural networks for certain GEPs for dimensionality reduction including CCA which means our method can be used for deep multiview representation learning. We demonstrate the effectiveness of our method for solving GEPs in the stochastic setting using canonical multiview datasets and demonstrate state-of-the-art performance for optimizing Deep CCA.

## 1 INTRODUCTION

A Generalised Eigenvalue Problem (GEP) is defined by two symmetric[1] matrices $A, B \in \mathbb{R}^{d \times d}$. They are usually characterised by the set of solutions to the equation

$$Aw = \lambda Bw \qquad (1)$$

with $\lambda \in \mathbb{R}, w \in \mathbb{R}^d$, called (generalised) eigenvalue and (generalised) eigenvector respectively. Note that by taking $B = I$ we recover the standard eigenvalue problem. We shall only be concerned with the case where $B$ is positive definite to avoid degeneracy; in this case one can find a basis of eigenvectors spanning $\mathbb{R}^d$. Without loss of generality, take $w_1, \ldots, w_d$ such a basis of eigenvectors, with decreasing corresponding eigenvalues $\lambda_1 \geq \cdots \geq \lambda_d$. The following variational characterisation Stewart & Sun (1990) provides a useful alternative, iterative definition: $w_k$ solves

$$\max_{w \in \mathbb{R}^d} w^\top A w \quad \text{subject to } w^\top B w = 1, \ w^\top B w_j = 0 \text{ for } j = 1, \ldots, k\text{-}1. \qquad (2)$$

There is also a simpler (non-iterative) variational characterisation for the top-$k$ subspace (that spanned by $\{w_1, \ldots, w_k\}$), namely

$$\max_{W \in \mathbb{R}^{d \times k}} \text{trace}(W^\top A W) \quad \text{subject to } W^\top B W = I_k \qquad (3)$$

again see Stewart & Sun (1990); the drawback of this characterisation is it only recovers the subspace and not the individual eigenvectors. We shall see that these two different characterisations lead to different algorithms for the GEP.

---

[1]or, more generally, Hermitian Stewart & Sun (1990)

Many classical dimensionality reduction methods can be viewed as GEPs including but not limited to Principal Components Analysis (Hotelling, 1933), Partial Least Squares Haenlein & Kaplan (2004), Fisher Discriminant Analysis Mika et al. (1999), and Canonical Correlation Analysis (CCA) (Hotelling, 1992).

Each of the problems above is defined at a population level, using population values of the matrices $A, B$, usually functionals of some appropriate covariance matrices. The practical challenge is the sample version: to estimate the population GEP where we only have estimates of $A, B$ through some finite number of samples $(z_n)_{n=1}^N$; classically, one just solves the GEP with $A, B$ estimated by plugging in the relevant sample covariance matrices.

However for very large datasets, the dimensionality of the associated GEPs makes it memory and compute intensive to compute solutions using existing full-batch algorithms; these are usually variants of the singular value decomposition where successive eigenvalue eigenvector pairs are calculated sequentially by deflation Mackey (2008) and so cannot exploit parallelism over the eigenvectors.

This work was motivated in particular by CCA, a classical method for learning representations of data with two or more distinct views: a problem known as multiview (representation) learning. Multiview learning methods are useful for learning representations of data with multiple sets of features, or 'views'. CCA identifies projections or subspaces in at least two different views that are highly correlated and can be used to generate robust low-dimensional representations for a downstream prediction task, to discover relationships between views, or to generate representations of a view that is missing at test time. CCA has been widely applied across a range of fields such as Neuroimaging (Krishnan et al., 2011), Finance (Cassel et al., 2000), and Imaging Genetics (Hansen et al., 2021).

Deep learning functional forms are often extremely effective for modelling extremely large datasets as they have more expressivity than linear models and scale better than kernel methods. While PCA has found a natural stochastic non-linear extension in the popular autoencoder architecture (Kramer, 1991), applications of Deep CCA (Andrew et al., 2013) have been more limited because estimation of the constraints in the problem outside the full batch setting are more challenging to optimize. In particular, DCCA performs badly when its objective is maximized using stochastic mini-batches. This is unfortunate as DCCA would appear to be well suited to a number of multiview machine learning applications as a number of successful deep multiview machine learning (Suzuki & Matsuo, 2022) and certain self-supervised learning approaches (Zbontar et al., 2021) are designed around similar principals to DCCA; to maximize the consensus between non-linear models of different views Nguyen & Wang (2020).

Recently, a number of algorithms have been proposed to approximate GEPs Arora et al. (2012), and CCA specifically Bhatia et al. (2018), in the 'stochastic' or 'data-streaming' setting; these can have big computational savings. Typically, the computational complexity of classical GEP algorithms is $\mathcal{O}\left((N+k)d^2\right)$; by exploiting parallelism (both between eigenvectors and between samples in a mini-batch), we can reduce this down to $\mathcal{O}(dk)$ (Arora et al., 2016). Stochastic algorithms also introduce a form of regularisation which can be very helpful in these high-dimensional settings.

A key motivation for us was a recent line of work reformulating top-k eigenvalue problems as games (Gemp et al., 2020; 2021), later extended to GEPs in Gemp et al. (2022). We shall refer to these ideas as the 'Eigengame framework'. Unfortunately, their GEP extension is very complicated, with 3 different hyperparameters; this complication is needed because they constrain their estimates to lie on the unit sphere, which is a natural geometry for the usual eigenvalue problem but not natural for the GEP. By replacing this unit sphere constraint with a Lagrange multiplier penalty, we obtain a much simpler method (GHA-GEP) with only a single hyperparameter; this is a big practical improvement because the convergence of the algorithms is mostly sensitive to step-size (learning rate) parameter Li & Jordan (2021), and it allows a practitioner to explore many more learning rates for the same computational budget. We also propose a second class of method ($\delta$-EigenGame) defined via optimising explicit utility functions, rather than being defined via updates, which enjoys the same practical advantages and similar performance. These utilities give unconstrained variational forms for GEPs that we have not seen elsewhere in the literature and may be of independent interest; their key practical advantage is that they only contain linear factors of $A, B$ so we can easily obtain unbiased updates for gradients. The other key advantage of these utility-based methods is that they

can easily be extended to use deep learning to solve problems motivated by GEPs. In particular we propose a simple but powerful method for the Deep CCA problem.

## 1.1 NOTATION

We have collected here some notational conventions which we think may provide a helpful reference for the reader. We shall always have $A, B \in \mathbb{R}^d$. We denote (estimates to or dummy variables for) the $i^{\text{th}}$ generalised eigenvectors by $w_i$; and denote CCA directions $u_i \in \mathbb{R}^p, v_i \in \mathbb{R}^q$. The number of directions we want to estimate will be $k$. For stochastic algorithms, we denote batch-size by $b$. We use $\langle \cdot, \cdot \rangle$ for inner products; implicitly we always take Euclidean inner product over vectors and Frobenius or 'trace' inner product for matrices.

## 2 A CONSTRAINT-FREE ALGORITHM FOR GEPs

Our first proposed method solves the general form of the generalized eigenvalue problem in equation (2) for the top-k eigenvalues and their associated eigenvectors in parallel. We are thus interested in both the top-k subspace problem and the top-k eigenvectors themselves. Our method extends the Generalized Hebbian Algorithm to GEPs, and we thus refer to it as GHA-GEP.

In the full-batch version of our algorithm, each eigenvector estimate has updates with the form

$$\Delta_i^{\text{GHA-GEP}} = \overbrace{A\hat{w}_i}^{\text{Reward}} - \overbrace{\sum_{j \leq i} B\hat{w}_j (\hat{w}_j^\top A\hat{w}_i)}^{\text{Penalty}} = \overbrace{A\hat{w}_i}^{\text{Reward}} - \overbrace{B\hat{w}_i(\hat{w}_i^\top A\hat{w}_i)}^{\text{Variance Penalty}} - \overbrace{\sum_{j < i} B\hat{w}_j (\hat{w}_j^\top A\hat{w}_i)}^{\text{Orthogonality Penalty}} \quad (4)$$

$$= \overbrace{A\hat{w}_i}^{\text{Reward}} - \overbrace{\sum_{j \leq i} B\hat{w}_j \Gamma_{ij}}^{\text{Penalty}} \quad (5)$$

where $\hat{w}_j$ is our estimator to the eigenvector associated with the $j^{\text{th}}$ largest eigenvalue and in the stochastic setting, we can replace $A$ and $B$ with their unbiased estimates $\hat{A}$ and $\hat{B}$. We will use the notation $\Gamma_{ij} = (\hat{w}_j^\top A\hat{w}_i)$ to facilitate comparison with previous work in Appendix A. $\Gamma_{ij}$ has a natural interpretation as Lagrange multiplier for the constraint $w_i^\top B w_j = 0$; indeed, Chen et al. (2019) prove that $(\hat{w}_j^\top A\hat{w}_i)$ is the optimal value of the corresponding Lagrange multiplier for their GEP formulation; we summarise this derivation in Appendix C.2 for ease of reference. We also label the terms as rewards and penalties to facilitate discussion with respect to the EigenGame framework in Appendix A.3 and recent work in self-supervised learning in Appendix E.

**Proposition 2.1** (Unique stationary point). *Given exact parents and assuming the top-k generalized eigenvalues of $A$ and $B$ are distinct and positive, the only stable stationary point of the iteration defined by (5) eigenvector $w_i$ (up to sign).*

## 2.1 DEFINING UTILITIES AND PSEUDO-UTILITIES WITH LAGRANGIAN FUNCTIONS

Now observe that our proposed updates can be written as the gradients of a Lagrangian pseudo-utility function:

$$\mathcal{PU}_i^{\text{GHA-GEP}}(w_i | w_{j<i}, \Gamma) = \tfrac{1}{2}\hat{w}_i^\top A\hat{w}_i + \tfrac{1}{2}\Gamma_{ii}(1 - \hat{w}_i^\top B\hat{w}_i) - \sum_{j<i} \Gamma_{ij}\hat{w}_j^\top B\hat{w}_i. \quad (6)$$

We show how this result is closely related to the pseudo-utility functions in Chen et al. (2019) and suggests an alternative pseudo-utility function for the work in Gemp et al. (2021) in Appendix C.3 which, unlike the original work, does not require stop gradient operators.

If we plug in the relevant $w_i$ and $w_j$ terms into $\Gamma$, we obtain the following utility function:

$$\mathcal{U}_i^\delta(w_i; w_{j<i}) = \tfrac{1}{2}\hat{w}_i^\top A\hat{w}_i + \tfrac{1}{2}\hat{w}_i^\top A\hat{w}_i\left(1 - \hat{w}_i^\top B\hat{w}_i\right) - \sum_{j<i}\hat{w}_i^\top A\hat{w}_j\hat{w}_j^\top B\hat{w}_i$$

$$= (\hat{w}_i^\top A\hat{w}_i) - \tfrac{1}{2}(\hat{w}_i^\top A\hat{w}_i)(\hat{w}_i^\top B\hat{w}_i) - \sum_{j<i}(\hat{w}_i^\top A\hat{w}_j)(\hat{w}_j^\top B\hat{w}_i) \qquad (7)$$

A remarkable fact is that this utility function actually defines a solution to the GEP problem! We prove the following consistency result in Appendix B.1.

**Proposition 2.2** (Unique stationary point). *Assuming the top-i generalized eigenvalues of the GEP (2) are positive and distinct. Then the unique maximizer of the utility in (7) for exact parents is precisely the $i^{th}$ eigenvector (up to sign).*

An immediate corollary is:

**Corollary 2.1.** *The top-k generalized eigenvectors form the unique, strict Nash equilibrium of $\Delta$-EigenGame*

Furthermore, the penalty terms in the utility function (6) have a natural interpretation as a projection deflation as shown in appendix C.5.

This utility function allows us to formalise $\Delta$-EigenGame, whose solution corresponds to the top-k solution of equation (2).

**Definition 2.1.** *Let $\Delta$-EigenGame be the game with players $i \in \{1, ..., k\}$, strategy space $\hat{w}_i \in \mathbb{R}^d$, where d is the dimensionality of A and B, and utilities $\mathcal{U}_i^\delta$ defined in equation (7*

Next note that it is easy to compute the derivative

$$\Delta_i^\delta = \frac{\partial \mathcal{U}_i^\delta(w_i; w_{j<i})}{\partial w_i} \qquad (8)$$

$$= 2A\hat{w}_i - \{A\hat{w}_i(\hat{w}_i^\top B\hat{w}_i) + (\hat{w}_i^\top A\hat{w}_i)B\hat{w}_i\} - \sum_{j<i}\{A\hat{w}_j(\hat{w}_j^\top B\hat{w}_i) + (\hat{w}_j^\top A\hat{w}_i)B\hat{w}_j\}$$

$$= \Delta_i^{\text{GHA-GEP}} + \{A\hat{w}_i - \sum_{j\leq i}A\hat{w}_j(\hat{w}_j^\top B\hat{w}_i)\}$$

This motivates an alternative algorithm for the GEP which we call $\delta$-EigenGame (where, consistent with previous work, we use upper case for the game and lower case for its associated algorithm).

## 2.2 STOCHASTIC/DATA-STREAMING VERSIONS

This paper is motivated by cases where the algorithm only has access to unbiased sample estimates of $A$ and $B$. These estimates, denoted $\hat{A}$ and $\hat{B}$, are therefore random variables. A nice property of both our proposed GHA-GEP and $\delta$-EigenGame is that $A$ and $B$ appear as multiplications in both of their updates (as opposed to as divisors). This means that we can simply substitute them for our unbiased estimates at each iteration. For the GHA-GEP algorithm this gives us updates based on stochastic unbiased estimates of the gradient

$$\hat{\Delta}_i^{\text{GHA-GEP}} = \hat{A}\hat{w}_i - \sum_{j\leq i}\hat{B}\hat{w}_j\left(\hat{w}_j^\top \hat{A}\hat{w}_i\right). \qquad (9)$$

Which we can use to form algorithm 1.

Likewise we can form stochastic updates for $\delta$-EigenGame

$$\Delta_i^\delta = 2\hat{A}\hat{w}_i - \{\hat{A}\hat{w}_i(\hat{w}_i^\top \hat{B}\hat{w}_i) + (\hat{w}_i^\top \hat{A}\hat{w}_i)\hat{B}\hat{w}_i\} - \sum_{j<i}\{\hat{A}\hat{w}_j(\hat{w}_j^\top \hat{B}\hat{w}_i) + (\hat{w}_j^\top \hat{A}\hat{w}_i)\hat{B}\hat{w}_j\} \quad (10)$$

Which give us algorithm 2.

Furthermore, the simplicity of the form of the updates means that, in contrast to previous work, our updates in the stochastic setting require only one hyperparameter - the learning rate.

---

**Algorithm 1** A Sample Based Generalized Hebbian Algorithm for GEP

---

**Input:** data stream $Z_t$ consisting of $b$ samples from $z_n$. Learning rate $(\eta_t)_t$. Number of time steps $T$. Number of eigenvectors to compute $k$.
**Initialise:** $(\hat{w}_i)_{i=1}^K$ with random uniform entries
**for** $t = 1$ **to** $T$ **do**
   Construct independent unbiased estimates $\hat{A}$ and $\hat{B}$ from $Z_t$
   **for** $i = 1$ **to** $k$ **do**
      $\hat{w}_i \leftarrow \hat{w}_i + \eta_t \hat{\Delta}_i^{\text{GHA-GEP}}$ {As defined in (9)}
   **end for**
**end for**

---

**Algorithm 2** A Sample Based $\delta$-EigenGame for GEP

---

**Input:** data stream $Z_t$ consisting of $b$ samples from $z_n$. Learning rate $(\eta_t)_t$. Number of time steps $T$. Number of eigenvectors to compute $k$.
**Initialise:** $(\hat{w}_i)_{i=1}^K$ with random uniform entries
**for** $t = 1$ **to** $T$ **do**
   Construct independent unbiased estimates $\hat{A}$ and $\hat{B}$ from $Z_t$
   **for** $i = 1$ **to** $k$ **do**
      $\hat{w}_i \leftarrow \hat{w}_i + \eta_t \Delta_i^{\delta}$ {As defined in (10)}
   **end for**
**end for**

---

### 2.3 COMPLEXITY AND IMPLEMENTATION

For the GEPs we are motivated by, and in particular for CCA, $\hat{A}$ and $\hat{B}$ are low rank matrices (specifically, they have at most rank $b$ where $b$ is the mini-batch size). This means that, like previous variants of EigenGame, our algorithm has a per-iteration cost of $\mathcal{O}(bdk^2)$. We can similarly leverage parallel computing in both the eigenvectors (players) and data to achieve a theoretical complexity of $\mathcal{O}(dk)$.

A particular benefit of our proposed form is that we only require one hyperparameter which makes hyperparameter tuning particularly efficient. This is particularly important as prior work has demonstrated that methods related to the stochastic power method are highly sensitive to the choice of learning rate Li & Jordan (2021). Indeed, by using a decaying learning rate the user can in principle run our algorithm just once to a desired accuracy given their computational budget. This is in contrast to recent work proposing an EigenGame solution to stochastic GEPs (Gemp et al., 2022) which requires three hyperparameters.

## 3 APPLICATION TO CCA AND EXTENSION TO FOR DEEP CCA

Previous EigenGame approaches have not been extended to include deep learning functions. Gemp et al. (2020) noted that the objectives of the players in $\alpha$-EigenGame were all generalized inner products which should extend to general function approximators. However, it was unclear how to translate the constraints in previous EigenGame approaches to the neural network setting. In contrast, we have shown that our work is constraint free but can still be written completely as generalized inner products for certain GEPs and, in particular, dimensionality reduction methods like CCA.

### 3.1 CANONICAL CORRELATION ANALYSIS

Suppose we have vector-valued random variables $X, Y \in \mathbb{R}^p, \mathbb{R}^q$ respectively. Then CCA (Hotelling, 1992) defines a sequence of pairs of 'canonical directions' $(u_i, v_i) \in \mathbb{R}^{p+q}$ by the iterative maximisations

$$\max_{u \in \mathbb{R}^p, v \in \mathbb{R}^q} \text{Cov}(u^\top X, v^\top Y) \quad \text{subject to } \text{Cov}(u^\top X) = \text{Cov}(v^\top Y) = 1, \tag{11}$$

$$\text{Cov}(u^\top X, u_j^\top X) = \text{Cov}(v^\top Y, v_j^\top Y) = 0 \text{ for } j < i.$$

Now write $\text{Cov}(X) = \Sigma_{XX}, \text{Cov}(Y) = \Sigma_{YY}, \text{Cov}(X,Y) = \Sigma_{XY}$. It is straightforward to show (Borga (1998)) that CCA corresponds to a GEP with

$$A = \begin{pmatrix} 0 & \Sigma_{XY} \\ \Sigma_{YX} & 0 \end{pmatrix}, \qquad B = \begin{pmatrix} \Sigma_{XX} & 0 \\ 0 & \Sigma_{YY} \end{pmatrix}, \qquad w = \begin{pmatrix} u \\ v \end{pmatrix}, \qquad d = p + q. \qquad (12)$$

For the sample version of CCA, suppose we have observations $(x_n, y_n)_{n=1}^N$, which have been pre-processed to have mean zero. Then the classical CCA estimator solves the GEV above with covariances replaced by sample covariances Anderson (2003). To define our algorithm in the stochastic case, suppose that at time step $t$ we define $\hat{A}_t, \hat{B}_t$ by plugging sample covariances of the mini-batch at time $t$.

### 3.2 $\delta$-EIGENGAME FOR CCA

We defined CCA by maximising correlation between linear functionals of the two views of data; we can extend this to DCCA by instead considering non-linear functionals defined by deep neural networks. Consider neural networks $f, g$ which respectively map $X$ and $Y$ to a $d$ dimensional subspace. We will refer to the $k^{\text{th}}$ dimension of these subspaces using $f_k(X)$ and $g_k(X)$ where $f(X) = [f_1(X), ..., f_d(X)]$ and $g(X) = [g_1(X), ..., g_d(X)]$. Deep CCA finds $f$ and $g$ which maximize $\text{Corr}(f_i(X), g_i(Y))$ subject to orthogonality constraints.

To motivate an algorithm, note that (7) is just a function of the inner products

$$\langle \hat{w}_i, A\hat{w}_j \rangle = \text{Cov}(u_i^\top X, v_j^\top Y) + \text{Cov}(v_i^\top Y, u_j^\top X)$$
$$\langle \hat{w}_i, B\hat{w}_j \rangle = \text{Cov}(u_i^\top X, u_j^\top X) + \text{Cov}(v_i^\top Y, v_j^\top Y)$$

So replacing $u_i^\top X$ with $f_i(X)$ and $v_i^\top Y$ with $g_i(Y)$, and using the short-hand

$$\tilde{A}_{ij} = \text{Cov}(f_i(X), g_j(Y)) + \text{Cov}(g_i(Y), f_j(X)) \qquad (13)$$
$$\tilde{B}_{ij} = \text{Cov}(f_i(X), f_j(X)) + \text{Cov}(g_i(Y), g_j(Y)) \qquad (14)$$

we obtain the objective

$$\mathcal{U}_i^\delta(f_i, g_i | f_{j<i}, g_{j<i}) = 2\tilde{A}_{ii} - \tilde{A}_{ii}\tilde{B}_{ii} - 2\sum_{j<i} \tilde{A}_{ij}\tilde{B}_{ij} \qquad (15)$$

Next observe by symmetry of matrices $\tilde{A}, \tilde{B}$ that if we sum the first $k$ utilities we obtain

$$\begin{aligned} U_k^{\text{sum}} = \sum_{i=1}^k U_i^\delta &= \sum_{i=1}^k 2\tilde{A}_{ii} - \sum_{i=1}^k \tilde{A}_{ii}\tilde{B}_{ii} - 2\sum_{i=1}^k \sum_{j<i} \tilde{A}_{ij}\tilde{B}_{ij} \\ &= 2\,\text{trace}(\tilde{A}) - \sum_{i,j=1}^k \tilde{A}_{ij}\tilde{B}_{ij} \\ &= 2\,\text{trace}(\tilde{A}) - \text{trace}(\tilde{A}\tilde{B}^\top) = \text{trace}\left(\tilde{A}(2I_k - \tilde{B})\right) \end{aligned} \qquad (16)$$

The key strength of this covariance based formulation is that we can obtain a full-batch algorithm by simply plugging in the sample covariance over the full batch; and obtain a mini-batch update by plugging in sample covariances on the mini-batch. We define DCCA-EigenGame in algorithm 3, where we slightly abuse notation: we write mini-batches in matrix form $X_t \in \mathbb{R}^{p \times b}, Y_t \in \mathbb{R}^{q \times b}$ and use short hand $f(X_t), f(Y_t)$ to denote applying $f, g$ to each sample in the mini-batch.

---

**Algorithm 3** DCCA EigenGame

---

**Input:** Stream of data with mini-batch size $b$ $\left(X_t \in \mathbb{R}^{b \times p}, Y_t \in \mathbb{R}^{b \times q}\right)$, neural networks $f(X)$, $g(Y)$ parameterized by $\hat{\theta}$ and $\hat{\psi}$, learning rate $\eta$
**for** $t = 1$ **to** $T$ **do**
    Construct unbiased estimates $\tilde{A}$ and $\tilde{B}$ from $f(X_t)$ and $g(Y_t)$
    $\mathcal{U} \leftarrow \text{trace}\left(\tilde{A}(2I_k - \tilde{B})\right)$
    $\tilde{\nabla}_f \leftarrow \frac{\partial \mathcal{U}}{\partial f}, \tilde{\nabla}_g \leftarrow \frac{\partial \mathcal{U}}{\partial g}$
    $\hat{\theta}_{t+1} \leftarrow \hat{\theta}_t + \eta\tilde{\nabla}_f, \hat{\psi}_{t+1} \leftarrow \hat{\psi}_t + \eta\tilde{\nabla}_g$
**end for**

---

We have motivated a loss function for SGD by a heuristic argument. We now give a theoretical result justifying the choice. Recall the top-$k$ variational characterisation of the GEP in (3) was hard to use in practice because of the constraints; we can use this to prove that the form above characterises the GEP.

**Proposition 3.1** (Subspace characterisation). *The top-$k$ subspace for the GEP (1) can be characterised by*

$$\max_{W \in \mathbb{R}^{d \times k}} \text{trace}\left(W^\top A W \left(2\,I_k - W^\top B W\right)\right) \tag{17}$$

We prove this result in Appendix B.3. We also provide an alternative derivation of the utility of (16) from the paper of Wang et al. (2015c) in Appendix C.1.

## 4 RELATED WORK

In particular we note the contemporaneous work in Gemp et al. (2022), termed $\gamma$-EigenGame, which directly addresses the stochastic GEP setting we have described in this work using an EigenGame-inspired approach. Since their method was designed around the Rayleigh quotient form of GEPs, it takes a different and more complicated form and requires additional hyperparameters in order to remove bias from the updates in the stochastic setting due to their proposed utility function containing random variables in denominator terms. It also isn't clear that their updates are the gradients of a utility function. Meng et al. (2021) developed an algorithm, termed RSG+, for streaming CCA which stochastically approximates the principal components of each view in order to approximate the top-k CCA problem, in effect transforming the data so that $B = I$ to simplify the problem. Arora et al. (2017) developed a Matrix Stochastic Gradient method for finding the top-k CCA subspace. However, the efficiency of this method depends on mini-batch samples of 1 and scales poorly to larger mini-batch sizes. While there have also been a number of approaches to the top-1 CCA problem (Li & Jordan, 2021; Bhatia et al., 2018), the closest methods in motivation and performance to our work on the linear problem are $\gamma$-EigenGame, SGHA, and RSG+.

The original DCCA (Andrew et al., 2013) was defined by the objective

$$\max \text{tracenorm}(\hat{\Sigma}_{XX}^{-1/2}\hat{\Sigma}_{XY}\hat{\Sigma}_{YY}^{-1/2}) \tag{18}$$

and demonstrated strong performance in multiview learning tasks when optimized with the full batch L-BFGS optimizer (Liu & Nocedal, 1989). However when the objective is evaluated for small mini-batches, the whitening matrices $\hat{\Sigma}_{XX}^{-1/2}$ and $\hat{\Sigma}_{YY}^{-1/2}$ are likely to be ill-conditioned, causing gradient estimation to be biased.

Wang et al. (2015b) observed that despite the biased gradients, the original DCCA objective could still be used in the stochastic setting for large enough mini-batches, a method referred to in the literature as stochastic optimization with large mini-batches (DCCA-STOL). Wang et al. (2015c) developed a method which adaptively approximated the covariance of the embedding for each view in order to whiten the targets of a regression in each view. This mean square error type loss can then be decoupled across samples in a method called non-linear orthogonal iterations (DCCA-NOI). To the best of our knowledge this method is the current state-of-the-art for DCCA optimisation using stochastic mini-batches.

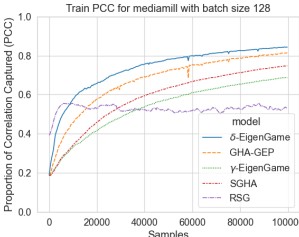 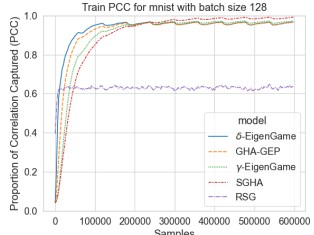 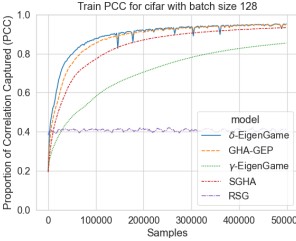

Figure 1: CCA with stochastic mini-batches: proportion of correlation captured with respect to Scipy ground truth b yGHA-GEP and $\delta$-EigenGame vs prior work. The maximum value is 1.

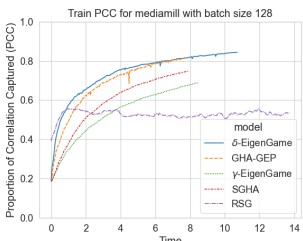 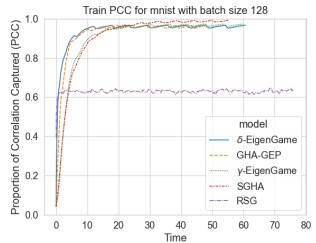 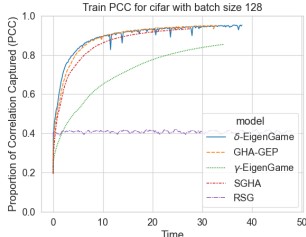

Figure 2: CCA with stochastic mini-batches: proportion of correlation captured with respect to Scipy ground truth by GHA-GEP and $\delta$-EigenGame vs prior work. The maximum value is 1.

## 5 EXPERIMENTS

In this section we replicate experiments from recent work on stochastic CCA and Deep CCA in order to demonstrate the accuracy and efficiency of our method.

### 5.1 STOCHASTIC SOLUTIONS TO CCA

In this section we compare GHA-GEP and $\delta$-EigenGame to previous methods for approximating CCA in the stochastic setting. We optimize for the top-8 eigenvectors for the MediaMill, Split MNIST and Split CIFAR datasets, replicating Gemp et al. (2022); Meng et al. (2021) with double the number of components and mini-batch size 128 and comparing our method to theirs. We use the Scipy (Virtanen et al., 2020) package to solve the population GEPs as a ground truth value and use the proportion of correlation captured (PCC) captured by the learnt subspace as compared to this population ground truth (defined in Appendix F.2).

Figure 1 shows that for all three datasets, both GHA-GEP and $\delta$-EigenGame exhibit faster convergence on both a per-iteration basis compared to prior work and likewise in terms of runtime in figure 2. They also demonstrate comparable or higher PCC at convergence. In these experiments $\delta$-EigenGame was found to outperform GHA-GEP. These results were broadly consistent across mini-batch sizes from 32 to 128 which we demonstrate in further experiments in Appendix G.1.

The strong performance of GHA-GEP and $\delta$-EigenGame is likely to be because their updates adaptively weight the objective and constraints of the problem and are not constrained arbitrarily to the unit sphere. We further explore the shape of the utility function in Appendix C.4.

### 5.2 STOCHASTIC SOLUTIONS TO DEEP CCA

In this section we compare DCCA-EigenGame and DCCA-SGHA to previous methods for optimizing DCCA in the stochastic setting. We replicated an experiment from Wang et al. (2015c) and compare our proposed methods to DCCA-NOI and DCCA-STOL. Like previous work, we use the total correlation captured (TCC) of the learnt subspace as a metric (defined in Appendix F.1).

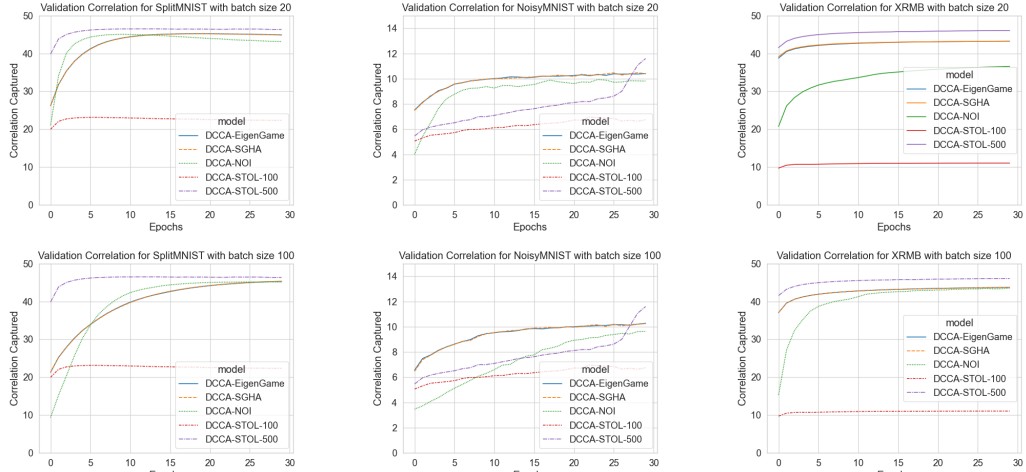

Figure 3: Total correlation captured by the 50 latent dimensions in the validation data. The maximum value is 50. The top row show results for mini-batch size 20 and the bottom row show results for mini-batch size 100

In all three datasets, figure 3 shows that DCCA-EigenGam finds higher correlations in the validation data than all methods except DCCA-STOL with $n = 500$ with typically faster convergence in early iterations compared to DCCA-NOI.

## 6 CONCLUSION

We have presented two novel algorithms for optimizing stochastic GEPs. The first, GHA-GEP was based on extending the popular GHA and we showed how it could be understood as optimising a Lagrangian psuedo-utility function. The second, $\delta$-EigenGame, was developed by swapping the Lagrange multipliers to give a proper utility function which allowed us to define the solution of a GEP with $\Delta$-EigenGame. Our proposed methods have simple and elegant forms and require only one choice of hyperparameter, making them extremely practical and both demonstrated comparable or better runtime and performance as compared to prior work.

We also showed how this approach can also be used to optimize Deep CCA and demonstrated state-of-the-art performance when using stochastic mini-batches. We believe that this will allow researchers to apply DCCA to a much wider range of problems.

In future work, we will apply $\delta$-EigenGame to other practically interesting GEPs like Generalized CCA for more than two views and Fisher Discriminant Analysis. We will also explore the extensions of other GEPs to the deep learning case in order to build principled deep representations.

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

## A    COMPARISON TO PREVIOUS WORK

### A.1    GENERALIZED HEBBIAN ALGORITHM

Our update is closely related to the Generalized Hebbian Algorithm (GHA) (Sanger, 1989) for solving the PCA problem with updates:

$$\Delta_i^{\text{GHA}} = A\hat{w}_i - \sum_{j \leq i} \hat{w}_j \left( \hat{w}_j^\top A \hat{w}_i \right) = A\hat{w}_i - \sum_{j \leq i} \hat{w}_j \Gamma_{ij} \tag{19}$$

which was originally designed to be solved sequentially rather than in parallel. Note that for GEPs where $B = I_d$ like PCA, our proposed method collapses exactly to GHA.

### A.2    STOCHASTIC GENERALIZED HEBBIAN ALGORITHM

To understand how our method extends GHA to generalized eigenvalue problems we consider the Stochastic Generalized Hebbian Algorithm (SGHA) (Chen et al., 2019)[2]. SGHA is derived from the min-max Lagrangian form of (3):

$$\min_{W \in \mathbb{R}^{d \times k}} \max_{\Gamma \in \mathbb{R}^{k \times k}} \mathcal{L}(W, \Gamma) = -\operatorname{tr}\left( W^\top A W \right) + \left\langle \Gamma, W^\top B W - I_k \right\rangle \tag{20}$$

Where $W$ is a matrix that captures the top-$k$ subspace (but not necessarily the top-k eigenvectors) and $\Gamma$ is a Lagrange multiplier that enforces the constraint in (2) along with the $B$-orthogonality of each eigenvector. By solving for the KKT conditions of equation (20), we have $\Gamma = \left( W^\top A W \right)$ and the authors propose to combine the primal and dual updates into a single step to give symmetrical updates for each eigenvector:

$$\Delta_i^{\text{SGHA}} = A\hat{w}_i - \sum_j B\hat{w}_j \left( \hat{w}_j^\top A \hat{w}_i \right) = A\hat{w}_i - \sum_j B\hat{w}_j \Gamma_{ij} \tag{21}$$

---

[2]Though the authors called this SGHA, it is rather different to the original proposal, because it is a subspace method rather than an iterative one

Where we highlight in red the key difference between our method and SGHA: that there is no hierarchy imposed on eigenvectors, so method can only recover top-$k$ subspace; this is in contrast to our proposal, all eigengame methods, and indeed the original GHA method. As noted by Gemp et al. (2020), imposing a hierarchy often appears to improve the stability of the algorithm in experiments and has the additional benefit of returning ordered eigenvectors.

### A.3 $\mu$-EigenGame

Finally, our method is closely related to $\mu$-EigenGame Gemp et al. (2021) - though this is only defined for the $B = I_d$ case. Their method restricts estimates to lie on the unit sphere, using Riemannian optimization tools to update in directions defined by

$$\Delta_i^\mu = A\hat{w}_i - \sum_{j<i} \hat{w}_j \big(\hat{w}_j^\top A\hat{w}_i\big) = A\hat{w}_i - \sum_{j<i} \hat{w}_j \Gamma_{ij} \tag{22}$$

where we again highlight in red the difference compared to our proposal: $\mu$-EigenGame however does not have the $j = i$ term in its penalty (and therefore does not use the $\Gamma_{ii}$ Lagrange multipliers associated with the unit variance constraint $\hat{w}_i^\top \hat{w}_i = 1$).

## B PROOFS AND FURTHER THEORETICAL ANALYSIS

### B.1 $\Delta$-EIGENGAME THEORY

We recall proposition 2.2:

**Proposition 2.2** (Unique stationary point). *Assuming the top-$i$ generalized eigenvalues of the GEP (2) are positive and distinct. Then the unique maximizer of the utility in (7) for exact parents is precisely the $i^{th}$ eigenvector (up to sign).*

*Proof.* For ease of reading the proofs in this appendix, we slightly change notation, and index the normalised solutions to the GEV with superscripts:

$$\langle w^{(i)}, Bw^{(i)}\rangle = 1, \quad \langle w^{(i)}, Aw^{(i)}\rangle = \lambda^{(i)} \,\forall i,$$

while we continue to index our estimates with subscripts. We can write our estimates in this basis to define the coefficients $\hat{w}_i = \sum_p \nu_i^{(p)} w^{(p)}$. Next define

$$m_i = \sum_p (\nu_i^{(p)})^2$$

$$z_i^{(j)} = \frac{(\nu_i^{(j)})^2}{\sum_p (\nu_i^{(p)})^2} = \frac{(\nu_i^{(j)})^2}{m_i}$$

so that the vector $z_i = (z_i^{(j)})_j$ takes values in the simplex. Then we have:

$$\langle \hat{w}_i, Aw^{(j)}\rangle = \lambda^{(j)} \nu_i^{(j)}$$
$$\langle \hat{w}_i, Bw^{(j)}\rangle = \nu_i^j$$
$$\langle \hat{w}_i, A\hat{w}_i\rangle = \sum_p \lambda^{(p)} (\nu_i^{(p)})^2$$
$$\langle \hat{w}_i, B\hat{w}_i\rangle = \sum_p (\nu_i^{(p)})^2 = m_i$$

Consider the utility function for player $i$:

$$u_i(\hat{w}_i|\hat{v}_{j<i}) = 2\langle \hat{w}_i, A\hat{w}_i \rangle - \langle \hat{w}_i, B\hat{w}_i \rangle \langle \hat{w}_i, A\hat{w}_i \rangle - 2\sum_{j<i}\langle \hat{w}_i, B\hat{w}_j \rangle \langle \hat{w}_j, A\hat{w}_i \rangle$$

$$= 2\sum_p \lambda^{(p)}(\nu_i^{(p)})^2 - 2\sum_{j<i}\lambda^{(j)}(\nu^j)^2 - (\sum_p \lambda^{(p)}(\nu_i^{(p)})^2)(\sum_p (\nu_i^{(p)})^2)$$

$$= 2\sum_{j \geq i}\lambda^{(j)}(\nu_i^{(j)})^2 - (\sum_p \lambda^{(p)}(\nu_i^{(p)})^2)(\sum_p (\nu_i^{(p)})^2)$$

$$= (2m_i - m_i^2)\sum_{j \geq i}\lambda^{(j)}z_i^{(j)} - m_i^2\sum_{j<i}\lambda^{(j)}z_i^{(j)}$$

Which is maximized when $m_i = 1$ $z_i^{(j)} = \delta_{ij}$, which implies that $\nu_i^i = \pm 1$ and $\nu_i^j = 0$ for $j \neq i$. $\qquad\square$

Note that in the previous lemma, the utility function took a simple form when we chose the true generalised eigenvectors as a basis; indeed, when using coefficients with respect to the basis, the utility only depended on the generalised eigenvalues and not the basis itself. This simple form shows how our utility interacts naturally with the geometry of the GEV problem. We will now analyse the corresponding simple form of the update steps.

## B.2 GHA-GEP Theory

We recall proposition 2.1:

**Proposition 2.1** (Unique stationary point). *Given exact parents and assuming the top-k generalized eigenvalues of $A$ and $B$ are distinct and positive, the only stable stationary point of the iteration defined by (5) eigenvector $w_i$ (up to sign).*

*Proof.* Let $w^{(j)}, \nu^{(j)}, \omega_i^j$ be defined as in the proof of proposition 2.2. Then the update

$$\Delta_i^\delta = A\hat{w}_i - B\hat{w}_i\left(\hat{w}_i^\top A\hat{w}_i\right) - \sum_{j<i}B\hat{w}_j\left(\hat{w}_j^\top A\hat{w}_i\right)$$

$$= A\hat{w}_i - \sum_{j \leq i}B\hat{w}_j\left(\hat{w}_j^\top A\hat{w}_i\right)$$

$$= \sum_p \lambda^{(p)}\nu_i^{(p)}Bw^{(p)} - \sum_{j \leq i}\left(\sum_p \nu_j^{(p)}Bw^{(p)}\right)\left(\sum_q \lambda^{(q)}\nu_i^{(q)}\nu_j^{(q)}\right)$$

$$= \sum_p Bw^{(p)}\left\{\lambda^{(p)}\nu_i^{(p)} - \sum_{j \leq i}\nu_j^{(p)}\left(\sum_q \lambda^{(q)}\nu_i^{(q)}\nu_j^{(q)}\right)\right\}$$

If we define the diagonal matrix $\Lambda = \mathrm{diag}(\{\lambda^{(p)}\}_p)$ and the vector $\nu_i = (\nu_i^{(p)})_{p=1}^d$ then we can rewrite:

$$\left(\sum_q \lambda^{(q)}\nu_i^{(q)}\nu_j^{(q)}\right) = \nu_i^\top \Lambda \nu_j.$$

Now by orthonormality of the $w^{(p)}$, we can equate coefficients and combine to vector form to obtain the update step for the coefficient vectors, which we shall notate by

$$\Delta(\nu)_i = \Lambda \nu_i - \sum_{j \leq i}\nu_j(\nu_j^\top \Lambda \nu_i)$$

$$= \left(I - \sum_{j \leq i}\nu_j\nu_j^\top\right)\Lambda\nu_i$$

$$= \left(I - \sum_{j<i}\nu_j\nu_j^\top\right)\Lambda\nu_i - \nu_i(\nu_i^\top \Lambda \nu_i) \qquad (23)$$

Only now do we consider the assumption of exact parents. This corresponds to the coefficient vectors $\nu_j = e_j \forall j < i$ where $e_j$ is the $j$th unit vector $((e_j)_k = \delta_{jk})$. Then

$$\left( I - \sum_{j<i} \nu_j \nu_j^\top \right) = \begin{pmatrix} 0_{(i-1)\times(i-1)} & 0_{(i-1)\times(i-1)} \\ 0_{(d-i+1)\times(i-1)} & I_{(d-i+1)\times(d-i+1)} \end{pmatrix}$$

So if we write $\bar{\lambda}_i = \nu_i^\top \Lambda \nu_i$ the update equations for our coefficients become:

$$\Delta(\nu)_i^{(p)} = -\bar{\lambda}_i \nu_i^{(p)} \quad \text{for } p < i \tag{24}$$

$$\Delta(\nu)_i^{(p)} = (\lambda^{(p)} - \bar{\lambda}_i)\nu_i^{(p)} \quad \text{for } p \geq i \tag{25}$$

So if we observe the qualitative behaviour:

- For $p < i$ the coefficients are shrunk towards zero (for sufficiently small step sizes)

- For $p \geq i$ the coefficients grow/decay depending on their generalised eigenvalue. The larger the eigenvalue, the more the components grow / the less they shrink. So over time only the $i$th component will be selected.

- The overall magnitude of the solution shrinks faster when $\bar{\lambda}_i$ is large.

A stationary point of the iteration therefore requires $\nu_i^{(p)} = 0$ for $p < i$. Then for each $p \geq i$ we must have either $\bar{\lambda}_i = \lambda^{(p)}$ or $\nu_i^{(p)} = 0$. Furthermore, $\nu_i^{(i)}$ grows at a faster rate than any of the other components, so provided this was non-zero at initialisation, it will be extracted uniquely. Finally, if $\nu_i^{(p)} = 0 \forall p \neq i$ but $\nu_i^{(i)} \neq 0$ then we must have $\lambda^{(i)} = \bar{\lambda}_i$ and also $\bar{\lambda}_i = \lambda^{(i)}(\nu_i^{(i)})^2$; so combining with we get $\nu_i^{(i)} = \pm 1$, as required. $\qquad\square$

### B.2.1 DISCUSSION OF CONTINUOUS DYNAMICS

In particular note that in the continuous time case above with exact parents, we can write the solutions to $\frac{d}{dt}\nu_i(t) = \Delta(\nu)_i$ with $\Delta(\nu)_i$ as in (24,25) as

$$\nu_i^{(p)}(t) = \nu_i^{(p)}(0) \exp\left( \mathbb{1}_{\{p \geq i\}} \lambda^{(p)} t - \int_{s=0}^t \bar{\lambda}_i(s)ds \right)$$

So when $z_i^{(i)}(0) \neq 0$ (hopefully this is almost sure), the trajectories on the simplex satisfy

$$\frac{z_i^{(j)}(t)}{z_i^{(i)}(t)} = \frac{z_i^{(j)}(0)}{z_i^{(i)}(0)} \exp\left( 2(\mathbb{1}_{\{p \geq i\}}\lambda^{(p)} - \lambda^{(i)})t \right) \to 0 \text{ as } t \to \infty$$

and we do indeed select the correct coefficient vector at an exponential rate. Note that in particular this equation for trajectory on the simplex is decoupled from the trajectory of the norm $m_i$ of the coefficients.

Of course, we are really interested in the case of in-exact parents. We can provide a heuristic argument similar to one of Gemp et al. (2021). Note that the updates for $w_i$ only depend on it's parents $w_{j<i}$, and one can show that an $\mathcal{O}(\epsilon)$ error in the parents propagates to an $\mathcal{O}(\epsilon)$ direction in the child gradient. We know $w_1$ will converge very fast to an arbitrary accuracy; then the gradient for $w_2$ will be very close to that corresponding to exact parents, so will quickly converge to that a similar order of accuracy; then the gradient of $w_3$ will be close to that for exact parents and so on.

### B.2.2 EXTENDING TO STOCHASTIC CASE

The real case of interest is the discrete time case with mini-batches. Gemp et al. (2021) claim that their algorithm converges almost surely provided the step-size sequence $\eta_t$ satisfies

$$\sum_{t=1}^\infty \eta_t = \infty, \qquad \sum_{t=1}^\infty \eta_t^2 < \infty \tag{26}$$

Their key tool is a result on Stochastic Approximation (SA) on Riemannian manifolds Shah (2017). This result extends the now-classical ODE method for analysis of SA schemes to Riemannian manifolds, mostly drawing on the presentation of Borkar (2008). One key difficulty of applying the literature on SA schemes is obtaining stability bounds (saying that the estimates never get too big); this becomes trivial when considering updates on compact manifolds like the unit sphere, which is why Gemp et al. (2021) are able to apply their SA tool 'out-of-the-box'. In our case, because we do not restrict to the unit sphere, we are able to apply more classical results on SA, for example Kushner et al. (2003), however, we would need to prove the corresponding stability estimates. These should hold intuitively because the variance penalty term should keep estimates small, but they are technically difficult. We note that obtaining such stability estimates has attracted a lot of theoretical attention, but in practice they are often unnecessary because only a bounded subset of the parameter space is physically sensible. This applies to our GEP: it only makes sense to consider vectors $w$ with $w^T B w \leq 1$; and though $B$ is unknown in general, we may well be able to lower bound its eigenvalues, giving a bounded parameter space of interest. We could modify our algorithm to project onto this bounded subset of parameter space. The theory of Kushner et al. (2003) can be applied to such a case of projected SA; indeed this theory has the added advantage of requiring weaker conditions on the step-sizes, namely only that

$$\sum_{t=1}^{\infty} \eta_t = \infty, \qquad \eta_t \to 0$$

Note such step-size schedules with slower decay is sometimes observed to give better empirical results in other SA problems Kushner et al. (2003).

We now point out what we understand to be a technical oversight in the proof of almost sure convergence in Gemp et al. (2021). They proof that $w_i$ converges a.s. to true value given fixed exact parents $w_{j<i}$ appears valid; as does the conclusion that $w_i$ converges a.s. to a corresponding optimum given fixed inexact parents; and also does their statement that if parents are close to correct then the corresponding optimum is close to correct. However, this does not say anything about convergence of $w_i$ when the parents are inexact and *varying*; in particular the arguments of Shah (2017) do not apply in this case. We believe that it would be possible to fix this oversight by considering a suitable coupling of solution paths starting from an $\epsilon$-covering of a neighbourhood of the true solution. Alternatively, it may be possible to apply the result of Shah (2017) to the combined estimates $(w_1, \ldots, w_k)$. Similar analysis will be needed for GEP-GHA because we also propose parallel updates for computational speed.

We next note that this SA literature also applies to our $\delta$-eigengame algorithm, whose updates are unbiased estimates to the gradient of the utilities $\mathcal{U}_i^{\delta}$. In this case, analysis may be more straightforward because we can also apply other existing literature on stochastic gradient descent.

We have not yet had time to make the discussion above more rigorous; we plan to do so in future work. Our algorithms fit very naturally into the well-studied SA framework, and we expect this literature to contain useful intuition and suggestions for implementation, as well as theoretical guarantees.

### B.3 SUBSPACE CHARACTERISATION

We recall proposition 3.1:

**Proposition 3.1** (Subspace characterisation). *The top-$k$ subspace for the GEP (1) can be characterised by*

$$\max_{W \in \mathbb{R}^{d \times k}} \text{trace} \left( W^\top A W \left( 2 I_k - W^\top B W \right) \right) \tag{17}$$

*Proof.* Let the objective function be $h(W)$. Firstly note that because $B$ is assumed positive definite, $h$ becomes negative for sufficiently large values of $W$, in particularly outside some compact set $C$. Second, note that for small $W$, $h$ is positive. Therefore any maximizer must be in $C$. Next note that $h$ is a composition of the trace (a linear map) with matrix polynomial so is differentiable, and in particular continuous; so the maximum on $C$ is attained. Further, at any maximizer $W$, the derivative $h'(W)$ is zero. We now compute it:

$$h'(W) = 2AW(W^\top B W - 2I) + 2BW(W^\top A W)$$

Setting to zero, left multiplying by $W^\top$, and using the previous notation $\tilde{A} = W^\top A W, \tilde{B} = W^\top B W$ gives

$$\tilde{B}\tilde{A} + \tilde{A}\tilde{B} = 2\tilde{A}$$

and so $(\tilde{B} - I)\tilde{A} + \tilde{A}(I - \tilde{B}) = 0$ so $S := (\tilde{B} - I)\tilde{A}$ is skew-symmetric (using that $A, B$ are both symmetric). But then right multiplying by $\tilde{A}^{-1}$ gives

$$\overbrace{\tilde{B} - I}^{\text{symmetric}} = \overbrace{S\tilde{A}^{-1}}^{\text{skew-symmetric}}$$

So in fact we must have both sides equal to zero and therefore $\tilde{B} = I$. But then by the arguments at the start of the proof, we see that any maximizer of $h$ must in fact have $W^\top B W = I$; so (17) is indeed equivalent to (3) and any maximizer recovers the top-$k$ subspace. $\square$

We now indulge in some vague intuition: a key strength of this unconstrained formulation is that it is straightforward to transform with respect to arbitrary changes of basis; therefore one can do analysis in the basis of generalised eigenvectors. By contrast, the orthogonality constraint in (3) only permits orthogonal changes of basis. This may give intuition to why $\mu$-eigengame only works in the $B = I$ case but our approach is effective for general GEPs.

## C  FURTHER CONNECTIONS TO PREVIOUS WORK

### C.1  RELATIONSHIP TO PREVIOUS DCCA FORMULATIONS

In Wang et al. (2015c), DCCA is formalised as

$$\max_{f,g,U \in \mathbb{R}^{d_x \times k}, V \in \mathbb{R}^{d_y \times k}} \text{trace}(U^\top F G^\top V) \quad \text{subject to } U^\top F F^\top U = V^\top G G^\top V = I_k$$

where $F = (f(x_1), \ldots, f(x_N)), G = (g(y_1), \ldots, g(y_n))$ are matrices whose columns are images of the training data under functions $f, g$ defined by neural networks in some class of functions $\mathcal{F}, \mathcal{G}$ with input and output dimensions $(p, d_x), (q, d_y)$ respectively. Observe that this optimisation is really targeting the population problem

$$\max_{f \in \mathcal{F}, g \in \mathcal{G}, U \in \mathbb{R}^{d_x \times k}, V \in \mathbb{R}^{d_y \times k}} \text{trace}(U^\top \Sigma_{fg} V) \quad \text{subject to } U^\top \Sigma_{ff} U = V^\top \Sigma_{gg}^\top V = I_k$$

Where $\Sigma_{ff} = \text{Var}(f(X)), \Sigma_{f,g} = \text{Cov}(f(X), g(Y)), \Sigma_{gg} = \text{Var}(g(Y))$. We now abuse notation to write $R_k(f(X), g(Y))$ to correspond to the sum of the first $k$ canonical correlations of the pair of random variables $f(X), g(Y)$. It is well known that if we fix $f, g$ in the above then the optimisation problem defines the top-$k$ subspace for CCA. So we can write the optimisation as

$$\max_{f \in \mathcal{F}, g \in \mathcal{G}} R_k(f(X), g(X))$$

But then using proposition 3.1, this optimisation is also equivalent to

$$\max_{f \in \mathcal{F}, g \in \mathcal{G}, W \in \mathbb{R}^{(d_x + d_y) \times k}} \text{trace}\left(W^\top A_{fg} W \left(2 I_k - W^\top B_{fg} W\right)\right) \tag{27}$$

where we define

$$A_{fg} = \begin{pmatrix} 0 & \Sigma_{fg} \\ \Sigma_{gf} & 0 \end{pmatrix}, \qquad B_{fg} = \begin{pmatrix} \Sigma_{ff} & 0 \\ 0 & \Sigma_{gg} \end{pmatrix}, \qquad d = p + q.$$

We have now almost recovered the form of (16), the only difference is that there is an optimisation over W in the above. To finish the derivation we follow Wang et al. (2015c) and define the augmented function classes:

$$\tilde{\mathcal{F}} = \{\tilde{f} = U^\top f : f \in \mathcal{F}, U \in \mathbb{R}^{d_x \times k}\}, \quad \tilde{\mathcal{G}} = \{\tilde{g} = V^\top g : g \in \mathcal{G}, V \in \mathbb{R}^{d_y \times k}\}$$

For any $W \in \mathbb{R}^{(d_x + d_y) \times k}$ there exist unique $U \in \mathbb{R}^{d_x \times k}, V \in \mathbb{R}^{d_y \times k}$ with $W^\top = (U^\top, V^\top)$. Also for $\tilde{f} = U^\top f, \tilde{g} = V^\top g$ it follows from definition of covariance that

$$U^\top \Sigma_{ff} U = \Sigma_{\tilde{f}\tilde{f}}, \quad U^\top \Sigma_{fg} U = \Sigma_{\tilde{f}\tilde{g}}, \quad V^\top \Sigma_{gg} V = \Sigma_{\tilde{g}\tilde{g}}$$

so indeed we can write (27) as

$$\max_{f \in \tilde{\mathcal{F}}, g \in \tilde{\mathcal{G}}} \text{trace} \left( A_{\tilde{f}\tilde{g}} \left( 2 I_k - B_{\tilde{f}\tilde{g}} \right) \right) \tag{28}$$

which precisely matches our objective in (16).

We now comment on this analysis: the definition from Wang et al. (2015c) proposes DCCA to find a pair of low-dimensional feature maps under which the two sets of data are highly correlated. Intuitively, this analysis says that if we take a sufficiently expressive class of neural networks, we only need to consider a $k$ dimensional latent space to recover the top-$k$ subspace of 'deep canonical directions'. Note also that one only needs to apply a $k$-dimensional classical CCA to recover the top-$k$ directions from this subspace. Finally, we warn that in general these directions may be highly non-unique, and that many of the nice properties of CCA are dependent on the structure of Euclidean space and do not hold for DCCA.

## C.2 DERIVATION OF SGHA ALGORITHM FROM CHEN ET AL. (2019)

The Lagrangian function in Chen et al. (2019) corresponding to (3) is given as:

$$\mathcal{L}(W, \Gamma) = \text{tr} \left( W^\top A W \right) - \left\langle \Gamma, W^\top B W - I \right\rangle \tag{29}$$

Differentiating with respect to $W$ gives

$$2AW - 2BW\Gamma = 0$$

Left multiplying by $W^T$ and using the constraint $W^T B W = I_k$ shows that at any stationary point we have

$$\Gamma = W^T A W \tag{30}$$

They then plug this value of $\Gamma$ into a gradient descent step for $W$ to obtain an update direction:

$$\Delta^{\text{SGHA}}(W) = -AW - BW(W^T A W)$$

(where we follow their exposition and drop the factor of 2 at this point). Note that this technique of plugging in the optimal dual variable is non-standard to our knowledge. The algorithm needs their theoretical results for more concrete justification.

## C.3 PSEUDO-UTILITIES IN PREVIOUS WORK

Recall we defined the pseudo-utility of GHA-GEP to be

$$\mathcal{PU}_i^{\text{GHA-GEP}}(w_i | w_{j<i}, \Gamma) = \hat{w}_i^\top A \hat{w}_i + \Gamma_{ii}(1 - \hat{w}_i^\top B \hat{w}_i) - 2 \sum_{j<i} \Gamma_{ij} \hat{w}_j^\top B \hat{w}_i \tag{31}$$

is closely related to expressions in previous work. Consider first SGHA. The Lagrangian function in Chen et al. (2019) is given above in (29).

Considering a single 'player' this utility can be written:

$$\mathcal{PU}_i^{\text{SGHA}}(w_i, \Gamma) = \hat{w}_i^\top A \hat{w}_i + \Gamma_{ii}(1 - \hat{w}_i^\top B \hat{w}_i) - \sum_j \Gamma_{ij} \hat{w}_j^\top B \hat{w}_i \tag{32}$$

We can also write the updates of $\mu$-EigenGame Gemp et al. (2021) as a Lagrangian pseudo-utility. Note that this is a slightly different expression to that given by the authors.

$$\mathcal{PU}_i^{\mu}(w_i | w_{j<i}, \Gamma) = \hat{w}_i^\top A \hat{w}_i - \sum_{j<i} \Gamma_{ij} \hat{w}_j^\top \hat{w}_i \tag{33}$$

## C.4 UTILITY SHAPE

**Lemma C.1.** *Let $\hat{w}_i = m(\cos(\theta_i) w_i + \sin(\theta_i) \Delta_i)$ where $\hat{w}_i^\top B \hat{w}_i = m$, then:*

$$\mathcal{U}_i(\hat{w}_i, w_{j<i}) = \mathcal{U}_i(mw_i, w_{j<i}) - \sin^2(\theta_i)(\mathcal{U}_i(mw_i, w_{j<i}) - \mathcal{U}_i(m\Delta_i, w_{j<i})) \tag{34}$$

We will show that this result follows from similar logic to Gemp et al. (2020) once the scaling factor $m$ is accounted for.

*Proof.* Let $\Delta_i = \sum_{l=1}^d p_l w_l, \|p\| = 1$. Decomposing the utility function for player $i$ we have:

$$\mathcal{U}_i(\hat{w}_i, w_{j<i}) = 2\langle \hat{w}_i, A\hat{w}_i \rangle - \langle \hat{w}_i, B\hat{w}_i \rangle\langle \hat{w}_i, A\hat{w}_i \rangle - 2\sum_{j<i}\langle \hat{w}_i, Bw^{(j)} \rangle\langle w^{(j)}, A\hat{w}_i \rangle \tag{35}$$

$$\begin{aligned}
&= (2m - m^2)(\cos^2(\theta_i)\lambda_{ii} + \sin^2(\theta_i)\langle \Delta_i, \lambda\Delta_i \rangle) \\
&\quad - 2m\sum_{j<i}\langle \cos(\theta_i) w_i + \sin(\theta_i)\Delta_i, Aw_j \rangle\langle \cos(\theta_i) w_i + \sin(\theta_i)\Delta_i, Bw_j \rangle
\end{aligned} \tag{36}$$

$$\begin{aligned}
&= (2m - m^2)(\cos^2(\theta_i)\lambda_{ii} + \sin^2(\theta_i)\langle \Delta_i, \lambda\Delta_i \rangle) \\
&\quad - 2m\sum_{j<i}\sin^2(\theta_i)\langle \Delta_i, Aw_j \rangle\langle \Delta_i, Bw_j \rangle
\end{aligned} \tag{37}$$

$$\begin{aligned}
&= (2m - m^2)\lambda_{ii} \\
&\quad - (2m - m^2)\sin^2(\theta_i)\lambda_{ii} + \sin^2(\theta_i)((2m - m^2)\langle \Delta_i, A\Delta_i \rangle \\
&\quad - 2m\sum_{j<i}\langle \Delta_i, Aw_j \rangle\langle \Delta_i, Bw_j \rangle
\end{aligned} \tag{38}$$

$$\begin{aligned}
&= (2m - m^2)\lambda_{ii} \\
&\quad - \sin^2(\theta_i)((2m - m^2)\lambda_{ii} + (2m - m^2)\langle \Delta_i, A\Delta_i \rangle - 2m\sum_{j<i}\langle \Delta_i, Aw_j \rangle\langle \Delta_i, Bw_j \rangle)
\end{aligned} \tag{39}$$

$$= u_i(mw_i, w_{j<i}) - \sin^2(\theta_i)(\mathcal{U}_i(mw_i, w_{j<i}) - \mathcal{U}_i(m\Delta_i, w_{j<i})) \tag{40}$$

$\square$

## C.5 UTILITY AS A PROJECTION DEFLATION

$$\mathcal{U}_i^\delta = 2\hat{w}_i^\top \overbrace{[I - \sum_{j<i} B\hat{w}_j\hat{w}_j^\top]}^{\text{projection deflation}} A\hat{w}_i - \hat{w}_i^\top B\hat{w}_i\hat{w}_i^\top A\hat{w}_i \tag{41}$$

Analogously to the previous work in $\alpha$-EigenGame (Gemp et al., 2020), the matrix $[I - \sum_{j\le i} B\hat{w}_j\hat{w}_j^\top]$ has a natural interpretation as a projection deflation.

## D   VECTORIZED $\delta$-EIGENGAME

---
**Algorithm 4** Vectorized $\delta$-EigenGame

---
    **Input:** data stream $Z_t$ consisting of $b$ samples from $z_n$, learning rate $\eta$
    **for** $t = 1$ **to** $T$ **do**
        Construct independent unbiased estimates $\hat{A}$ and $\hat{B}$ from $Z_t$
        Rewards $\leftarrow 2A\hat{W}$
        Penalties $\leftarrow B\hat{W} \operatorname{triu}(\hat{W}^\top A\hat{W})$
        $\tilde{\nabla} \leftarrow$ Rewards - Penalties
        $\hat{W}' \leftarrow \hat{W} + \eta_t \tilde{\nabla}$
    **end for**

---

Where $\operatorname{triu}$ returns a matrix with the entries below the main diagonal set to zero.

## E   A CONNECTION TO SELF-SUPERVISED LEARNING METHODS

Reorganizing our update equation (5), we find that the intuition of our method can also be understood as three terms encouraging variance in $A$, penalizing variance in $B$, and discouraging covariance.

$$\Delta_i^\delta = \overbrace{A\hat{w}_i}^{\text{Reward}} - \overbrace{B\hat{w}_i\left(\hat{w}_i^\top A\hat{w}_i\right)}^{\text{Variance Penalty}} - \overbrace{\sum_{j<i} B\hat{w}_j\left(\hat{w}_j^\top A\hat{w}_i\right)}^{\text{Orthogonality Penalty}} \tag{42}$$

The motivation is similar to that in recent work in self-supervised learning (Zbontar et al., 2021) and in particular the VICReg method in Bardes et al. (2021). Recent work has shown links between several self-supervised learning approaches and classical spectral embedding methods (Balestriero & LeCun, 2022), some of which could be represented by GEPs. Like CCA, many self-supervised learning approaches are based on finding a function which is invariant to an image and its augmented version i.e. the learnt representations of both are correlated.

## F   EXPERIMENT DETAILS

### F.1   TOTAL CORRELATION CAPTURED (TCC)

This is the sum of the canonical correlations of the learnt representation (i.e. the sum of the top-k canonical correlations of $X$ and $Y$).

### F.2   PROPORTION OF CORRELATION CAPTURED (PCC)

This is the sum of the canonical correlations of the learnt representation as a proportion of the sum of the canonical correlations of the learnt representation using the population ground truth (i.e. the sum of the top-k canonical correlations of $X$ and $Y$).

### F.3   STOCHASTIC CCA

The latter two datasets are formed from left and right halves of the canonical datasets (LeCun et al., 2010; Krizhevsky et al., 2009). With the same initialization for all methods, we trained for 10 epochs on each dataset with a mini-batch size of 128 and illustrate the models with the best performance in the validation set.

### F.4   STOCHASTIC CCA HYPERPARAMETERS

Learning rate was tuned from $\eta = (10^{-1}, 10^{-2}, 10^{-3}, 10^{-4}, 10^{-5})$ and $\gamma$-EigenGame parameter $\gamma$ was tuned from the same range. We used Jax (Babuschkin et al., 2020) to optimize the linear CCA

models using the Jaxline framework. We used WandB (Biewald, 2020) for experiment tracking to develop insights for this paper.

### F.5 DEEP CCA

We use two variants of paired MNIST datasets. The first is identical to the split MNIST dataset in the previous section. The second harder problem is closely related to an experiment by Wang et al. (2015a). Their 'noisy' paired MNIST data takes two different digits with the same class. The first is rotated randomly while the second has additive gaussian noise. Finally, we use the X-Ray Microbeam (XRMB) dataset from Arora et al. (2016). For all of the datasets, we use two encoders with 50 latent dimensions and two hidden layers with size 800 and leaky ReLu activation functions, a similar architecture to that used in Wang et al. (2015c). We compare our proposed method to DCCA-NOI at mini-batch sizes of 20 and 100 and DCCA-STOL with mini-batch size 100 and 500 (DCCA-STOL cannot be used for mini-batch sizes less than the number of latent dimensions).

### F.6 DCCA HYPERPARAMETERS

We trained for 30 epochs on each dataset with mini-batch sizes of 20 and 100. Learning rate was tuned from $\eta = (10^{-1}, 10^{-2}, 10^{-3}, 10^{-4}, 10^{-5})$ and the DCCA-NOI parameter $\rho$ was tuned between 0 and 1. We use PyTorch (Paszke et al., 2019) with the Adam optimizer (Kingma & Ba, 2014).

## G ADDITIONAL EXPERIMENTS

### G.1 STOCHASTIC CCA WITH SMALLER MINI-BATCH SIZES

In this section we repeat the experiments described in the main text with smaller mini-batch sizes (64 and 32).

Results for mini-batch sizes 32 and 64 are broadly similar to those in the main text for mini-batch size 128. In the MNIST data we can see again that there is a tradeoff between speed of convergence in early iterations and the quality of the solution.

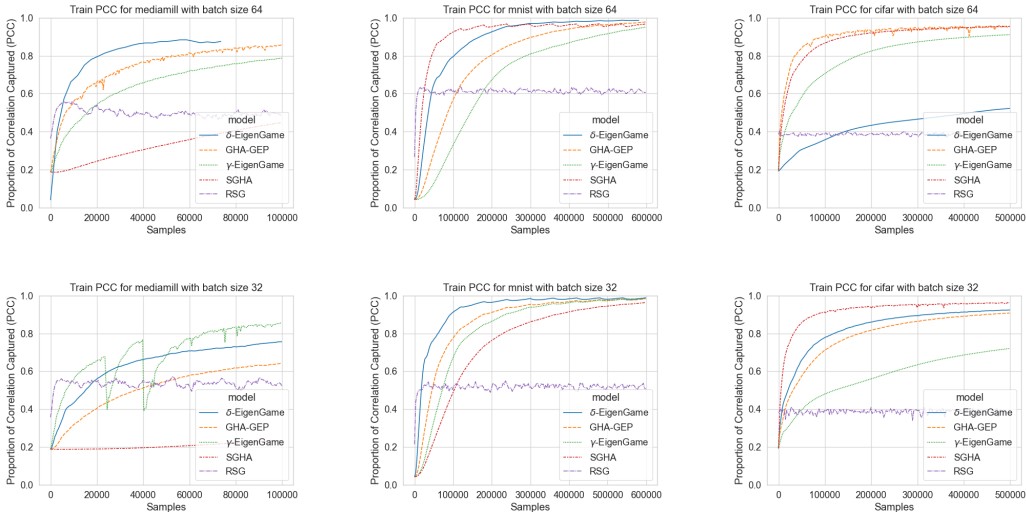

Figure 4: CCA with stochastic mini-batches of size 64 (top) and 32 (bottom): proportion of correlation captured with respect to Scipy ground truth by $\delta$-EigenGame vs prior work. The maximum value is 1.

## G.2 PARTIAL LEAST SQUARES

The Partial Least Squares (PLS) Wold et al. (1984) problem can also be formulated as a similar GEP but with $B$ replaced by the identity matrix. PLS is equivalent to finding the singular value decomposition (SVD) of the covariance matrix $X^\top Y$. It has an interpretation as a (infinitely) ridge regularised CCA where the covariance matrices $\Sigma_{XX}$ and $\Sigma_{YY}$ are replaced by identity matrices; this corresponds to assuming no collinearity between variables.

### G.2.1 PLS WITH STOCHASTIC MINI-BATCHES

In this experiment we compare our method to the Stochastic Power method Arora et al. (2016), $\gamma$-EigenGame, and SGHA for the stochastic PLS problem.

For these experiments we use the Proportion of Variance captured (PV). This is the sum of the singular values of the learnt representation using each stochastic optimisation method as a proportion of the sum of the singular values of the learnt representation using the population ground truth (i.e. the sum of the top-k singular values of the covariance matrix $X^\top Y$).

Figure 5 shows that all of the methods perform similarly in terms of variance captured across the datasets. While the stochastic power method is very fast to converge in the MNIST and CIFAR data, it solutions can be suboptimal. The performance of $\delta$-EigenGame is arguably more suprising for the PLS problem because both the Stochastic Power method and $\gamma$-EigenGame explicitly enforce the constraints at each iteration whereas $\delta$-EigenGame only enforces the constraint via penalty terms.

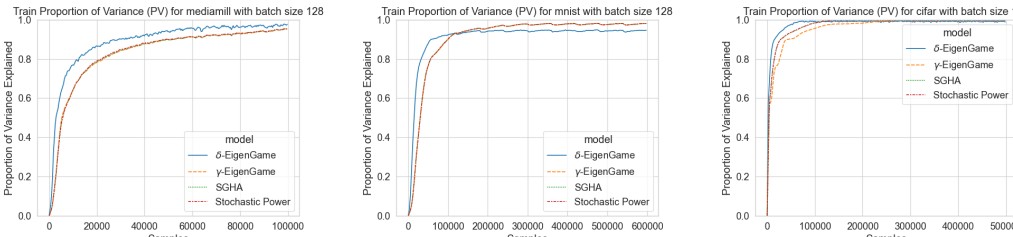

Figure 5: PLS with stochastic mini-batches: proportion of variance captured with respect to Scipy ground truth by $\delta$-EigenGame vs prior work. The maximum value is 1.

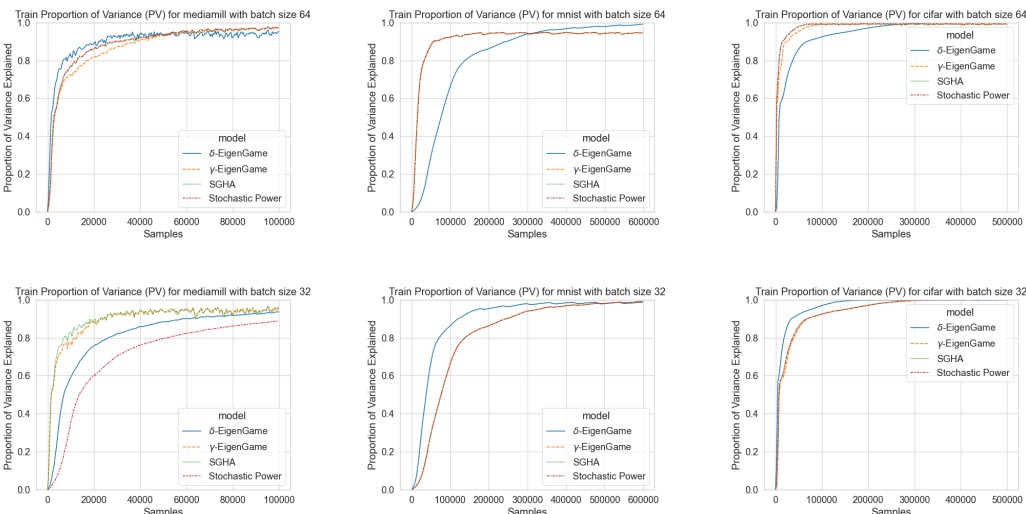

Figure 6: PLS with stochastic mini-batches of size 64 (top) and 32 (bottom): proportion of variance captured with respect to Scipy ground truth by $\delta$-EigenGame vs prior work. The maximum value is 1.

