# OpenReview forum: "A Generalized EigenGame With Extensions to Deep Multiview Representation Learning"
_ICLR.cc/2023/Conference — Submitted to ICLR 2023_

### Official Review · Reviewer_sgfU · 2022-10-24

**Confidence:** 4
**Clarity, Quality, Novelty And Reproducibility:** All good.
**Correctness:** 3
**Technical Novelty And Significance:** 2
**Empirical Novelty And Significance:** 2
**Recommendation:** 3

**Strength And Weaknesses:**


Strengths:
-) The discussion of related work is sufficient.
-) Numerical experiments confirm that the method is competitive from a qualitative perspective.
-) The proposed scheme appears to be well-fit for parallel computing, although this direction is not explored in the paper.

Weaknesses:
-) The paper is poorly written and some concepts, like the ones presented in Section 2 before Section 2.1 are not explained well (also, Eq. (13) is empty). Eqs (16), (17) etc. are poorly motivated, and it is not clear how they appear.
-) The novelty of the algorithm seems to lack. On several occasions, the paper feels to step on the work by Gemp et al. and the authors try hard to overstate the potential differences; to the point that it disrupts the natural flow of the submission.
-) Unless I missed it, there are no running time comparisons between different schemes. Note that faster convergence does not translate to a better algorithm if there are no running time advantages (or maximum qualitative accuracy captrured).

Minor comments:
-) The \hat operators below Eq. (15) are misaligned.

**Summary Of The Paper:**

This paper presents a top-k eigenvalue solver for stochastic eigenvalue problems. The proposed approach undertakes an optimization viewpoint using Lagrange multipliers. The effectiveness of the proposed scheme is validated through numerical experiments on simulated datasets.

**Summary Of The Review:**

My main concerns are: a) novelty, and  b) advantage over existing methods since no running-times are presented.

---

> ### Author Response · Authors · 2022-11-18
> **Response to Reviewer sgfU**
>
> We would like to thank the reviewer for their time and valuable comments. We have addressed reviewer sgfU’s comments on novelty and advantages over existing work in the general response to all reviewers’ as well as in the updated manuscript. Below we provide a response to reviewer sgfU’s specific comments:
>
> > *The paper is poorly written and some concepts, like the ones presented in Section 2 before Section 2.1 are not explained well (also, Eq. (13) is empty). Eqs (16), (17) etc. are poorly motivated, and it is not clear how they appear.*
>
> We have increased the clarity of the paper throughout and we hope that reviewer sgfU recognises these improvements.
>
> > *The novelty of the algorithm seems to lack. On several occasions, the paper feels to step on the work by Gemp et al. and the authors try hard to overstate the potential differences; to the point that it disrupts the natural flow of the submission.*
>
> We agree with reviewer sgfU that we had disrupted the flow of the submission with comparisons to previous work and have edited our manuscript accordingly. In particular we have pushed the more detailed comparisons with previous work to the appendix and focussed much more on the detail of our approach.
>
> We believe the latest version more powerfully communicates the advantages of our method. We add that Gemp et al is considered contemporaneous by ICLR and we are not obligated to compare with their work. We chose to compare our method because we strongly believe in the benefits of our approach; our proposed method is much simpler to optimise (1 vs. 3 hyperparameters), it more clearly relates to prior work (EigenGame and GHA), and extends neatly (and uniquely among related work) to Deep CCA. We hope that our improvements to the paper highlight more clearly these advantages.
>
> > *Unless I missed it, there are no running time comparisons between different schemes. Note that faster convergence does not translate to a better algorithm if there are no running time advantages (or maximum qualitative accuracy captured).*
>
> We have added running time comparisons to the stochastic CCA experiments and note that our proposed methods compare favorably. We do not give running time comparisons for the Deep CCA experiments as the calculation of the utility (loss) is not the bottleneck for running time.

---

### Official Review · Reviewer_CZPK · 2022-10-27

**Confidence:** 4
**Correctness:** 3
**Technical Novelty And Significance:** 2
**Empirical Novelty And Significance:** 2
**Recommendation:** 5

**Clarity, Quality, Novelty And Reproducibility:**

The paper could be better written in the experimental section.
To be more precise, even if I am theorist, I would vote positively for an experimental mainly paper if the majority of the main draft has been devoted to explain  (i) the validity & generality of the experimental setup (ii) the intuition between the advantages of the model.

About the novelty, I expect to earn more intuition about the importance of the application in Deep Learning framework. I invite the authors to be more explanatory of the improvements and the necessity of Deep CCA and the potential impact of an ADMM optimization-style for EigenGame.



**Details Of Ethics Concerns:**

Non-applicable

**Strength And Weaknesses:**

This question falls within a general research thread that has recently attracted considerable attention in the literature. To start with a positive point about the current submission, the current authors proved that, indeed, the unique equilibrium of the new game is again the solution of the Gen. EigenProblem. In other words, they proved that the reduction is meaningful . I will return to this general positioning later. The main contribution of the paper is to explain that experimentally a Lagrangian Multipliers usage in the optimization process can accelerate the Game Dynamics Process.

An objection that I raise collectively with the ''EigenGame'' reductions for all versions, is the inclusion of the stop-differentiate operator, introduced by Gemp et al from the alpha-version. Under this way, we lose completely the ''game-theoretic'' intuition. Actually, it would be interesting to understand even if the utility function is smooth/continuous. An important weakness of this series of work is the finite sample complexity. See the discussion of the discussion at the initial work of \alpha-EigenGame.

**Summary Of The Paper:**

This paper studies the application of EigenGame formulation to solve the Generalized Eigenvalue Problem (GEPs). The previous work had defined the implementation $\alpha$ and $\mu$ - EigenGame, while this one defines the $\delta$- EigenGame version. EigenGame is a reduction of the problem to compute eigenvalues to the problem of computing a Nash equilibrium in a ''quasi''-collaborative/potential game.
In the previous line of work, in order to compute this strict-pure Nash equilibrium, a Riemannian Gradient Descent is employed, (as an analogue of Best-Response in a potential game). This work describes an alternative description of the game, which implicitly incorporates the constrained optimization via Lagrange multipliers.

**Summary Of The Review:**

Currently, I feel that the work is below the threshold. However, at the first glance, it is clear that the paper provides an interesting implementation details about simplifying difficulties of previous versions of EigenGames. However, as standalone work, it seems a bit incremental  improvement. In the case that authors would not like to focus on the theoretical aspect of the optimization (which is fare), it is necessary to explain the importance of the applications and the burdens that are overpassed thanks to this EigenGame version.

---

> ### Author Response · Authors · 2022-11-18
> **Response to Reviewer CZPK**
>
> We would like to thank the reviewer for their time and valuable comments. We have addressed reviewer CZPK’s comments on the necessity of Deep CCA and improvements on existing work in the general response to all reviewers’ as well as in the updated manuscript. Below we provide a response to reviewer CZPK’s specific comments:
>
> > *An objection that I raise collectively with the ''EigenGame'' reductions for all versions, is the inclusion of the stop-differentiate operator, introduced by Gemp et al from the alpha-version. Under this way, we lose completely the ''game-theoretic'' intuition. Actually, it would be interesting to understand even if the utility function is smooth/continuous.*
>
> We believe that reviewer CZPK will find our new description of the pseudo-utility function for GHA-GEP (equation 6) and utility function for $\delta$-EigenGame (equation 7) satisfying. In particular, this feedback has encouraged us to be more precise and ultimately led us to realize that we have in fact contributed two slightly different algorithms; GHA-GEP and $\delta$-EigenGame. Our use of Lagrange multipliers allows us to write the pseudo-utility function without using the stop-differentiate operator and we explore how previous EigenGames and even the GHA can be understood through this lens in Appendix D.
>
> > *I expect to earn more intuition about the importance of the application in Deep Learning framework. I invite the authors to be more explanatory of the improvements and the necessity of Deep CCA*
>
> We acknowledge that the importance of deep multiview representation learning was not clearly articulated and have made improvements to the manuscript to remedy this. We now provide more references to justify this claim. Deep CCA in particular has the potential to be a powerful tool in deep multiview learning because CCA captures precisely what is commonly called the consensus principle (that representations of distinct views should be similar to each other). Its limited use thus far can be attributed primarily to the fact that DCCA cannot be optimised easily in the stochastic setting. Notwithstanding the improved performance of stochastic methods, the memory requirements when modelling memory intensive modalities like Medical Images or Genetics, hinder the use of full-batch Deep CCA, particularly when using Hardware accelerators like GPUs. For this reason we strongly believe the impact of our work on the DCCA problem will be substantial.
>
> > *[I invite the authors to be more explanatory of] the potential impact of an ADMM optimization-style for EigenGame.*
> We have also explored the theory underlying our empirical results for the ADMM-style algorithm (GHA-GEP) in much more detail, particularly in the Appendix.
>
> However it is also worth noting that we have now separated out our contributions into an ADMM optimization style algorithm (GHA-GEP) and a utility function based form ($\delta$-EigenGame), and that it is the utility function based form that we use to motivate our Deep CCA algorithm, one answer is that we think the impact of the utility based algorithm has the greater potential for impact. Indeed, we believe that it opens the door to using deep learning to solve problems motivated by classical GEPs.

---

> > ### Comment · Reviewer_CZPK · 2022-11-20
> > **Acknowledgement of Response**
> >
> > I sincerely thank the authors for their effort. I should admit that their response had high quality which is something that I admire in this process. Since I am not convinced yet about the novelty, I would like to ask a simple question to the authors: What do you believe that it is the most important contribution of this work in comparison with the literature. At the end of the day, this is the most important question to be answered to examine if a publication achieves to pass the bar of the standards

---

> > > ### Author Response · Authors · 2022-11-21
> > > **Follow up to CZPK**
> > >
> > > We appreciate the kind words.
> > >
> > > We believe our most important contribution is our use of **unconstrained variational characterizations** to obtain simpler algorithms for solving GEPs in the **stochastic/streaming** setting and solving deep representation learning problems (specifically DCCA) inspired by GEPs.
> > >
> > > With respect to the literature:
> > >
> > > - Gemp 2022 use a different form for GEPs (the Rayleigh quotient form) and this means that it is non-trivial to estimate unbiased stochastic gradients with respect to their utility function (4). Specifically, as they note, the random variable B appears as a denominator whereas in our forms (6) and (7) - A and B can simply be substituted for unbiased stochastic estimates of A and B. This is why ultimately the form of our algorithm for the stochastic case is simpler and has fewer hyperparameters. **We believe our work is clearly similarly motivated but has unique qualities and opens up options for further theoretical analysis (discussed in appendix B)**.
> > > - We certainly build on the work of Chen 2019 but we additionally adopt the hierarchical form from the EigenGame line of work and go one step forwards in moving from our equation (4) to equation (7) and removing the Lagrange multipliers altogether. Our experiments suggest, consistent with Gemp 2020, that adding this hierarchy improves the convergence of the algorithm. By removing the Lagrange multipliers we are able to further develop the method for deep learning problems.
> > > - To the best of our knowledge our work is the first to extend ideas from stochastic GEPs in this way to the deep learning setting. We demonstrate state-of-the-art for stochastic Deep CCA but there is no reason why the ideas could not be applied to other similar problems (Fisher Discriminant Analysis being perhaps the most obvious).
> > >
> > > We provide a complete picture of how our work is related to previous work in appendices A and C.

---

### Official Review · Reviewer_nriX · 2022-10-28

**Confidence:** 2
**Clarity, Quality, Novelty And Reproducibility:** Difficult to read, and little differe…
**Correctness:** 3
**Technical Novelty And Significance:** 2
**Empirical Novelty And Significance:** 2
**Recommendation:** 3

**Strength And Weaknesses:**

While the proposed algorithm seems to be promising empirically, its description and motivations are unclear and under-developed in the manuscript.

The key elements that are presented to motivate the algorithm are its lack of hard constraints, which are supposedly "softly" enforced by Lagrange multipliers, and reduced number of hyperparameters. Yet, in the presentation of the algorithm, I see no description of Lagrange multipliers, or soft-enforcement of constraints, and at least for the full-batch version, Algorithm 1 in [Gemp et al](https://arxiv.org/abs/2206.04993) seems to have only the step-size as a hyper-parameter, just like the present paper, and the projection on the sphere doesn't seem like a big issue in practice.

Overall, I encourage the authors to provide a clearer and more formal description of the proposed algorithm, and its motivation (ideally in a way that doesn't involve stop-gradients as currently done in eq.(12)), and a precise comparison to previous work.

**Summary Of The Paper:**

The paper proposes new algorithms for CCA and Partial Least Squares problems based on game-theoretic formulations, following previous work by Gemp et al. The proposed algorithm relaxes some constraints and works well empirically.

**Summary Of The Review:**

The paper is not ready for publication in its current state. I encourage the authors to state the motivations more clearly and improve the presentation of the proposed method.

---

> ### Author Response · Authors · 2022-11-18
> **Response to Reviewer nriX**
>
> We would like to thank the reviewer for their time and valuable comments. We have addressed reviewer nriX’s comments on motivation and formal description of the algorithm in the general response to all reviewers’ as well as in the updated manuscript. Below we provide a response to reviewer nriX’s specific comments:
>
> > *The key elements that are presented to motivate the algorithm are its lack of hard constraints, which are supposedly "softly" enforced by Lagrange multipliers, and reduced number of hyperparameters. Yet, in the presentation of the algorithm, I see no description of Lagrange multipliers, or soft-enforcement of constraints*
>
> We have now highlighted the Lagrange multipliers throughout the description of the algorithm in blue (and additionally we show that prior work in the area can also be understood through this lens). Constraints are softly enforced because penalty terms are used to encourage the constraints to be fulfilled at convergence. We use overbraces in equation (4) to illustrate which terms form the objective and which terms penalise violation of the constraints.
>
> > *at least for the full-batch version, Algorithm 1 in Gemp et al seems to have only the step-size as a hyper-parameter, just like the present paper*
>
> This work (and indeed the work in Gemp et al) is not motivated by the full batch case (a point we have made efforts to make clearer in the introduction) so we feel that comparison with their full batch algorithm is interesting since we approach the problem from different angles ie they approach the Rayleigh quotient form of GEPs.
>
> > *the projection on the sphere doesn't seem like a big issue in practice.*
>
> While we agree with reviewer nriX that the riemannian gradient descent on the unit sphere employed by previous work is not an issue for the standard linear GEPs studied in this work, this property is precisely what prevents previous work extending to deep learning functions (as noted by the authors in appendix K.1 and I of Gemp 2020).
>
> We would like to thank reviewer nriX for their encouragement with respect to the promising empirical results.

---

### Author Response · Authors · 2022-11-18
**General Response and Summary of Changes to the Manuscript**

# Introduction

We would like to thank the reviewers for their efforts reviewing the paper. We appreciate that the reviewers unanimously noticed the strong empirical results in the paper and therefore recognised the promise in the work.

All three reviews contained important criticisms and we have put a lot of work into improving the paper with them in mind. It is clear that common to all the reviews were three key elements:

- unclear description of the algorithm (including stop gradient notation)
- improvement with respect to previous work
- unclear motivations and importance of the applications

In this response we would like to highlight the changes we have made to the text with respect to these concerns as well as some exciting theoretical developments we have made which we believe will be of interest to all of the reviewers.

# 1) Description of the algorithm (as well as removing stop gradient operators)
We have clarified that our work actually constitutes two different algorithms. We now refer to these as ‘GHA-GEP’ and ‘$\delta$-EigenGame’ and we specifically extend $\delta$-EigenGame to Deep CCA.

GHA-GEP is a direct extension of the Generalized Hebbian Algorithm (GHA) and we show that it can be understood through the lens of Lagrange multipliers. We now highlight these Lagrange multipliers in blue to stress them in response to reviewer feedback as firstly they are critical in allowing us to remove stop gradients from our description of the utility function of GHA-GEP and secondly they facilitate theoretical comparison with previous work.

By integrating the update of GHA-GEP, we obtain a pseudo-utility function whose optimum is provably the solution to the top-k GEP (NOTE: Lagrange multipliers remove the need for stop gradients in the description of the utility function). By replacing the Lagrange multipliers in this pseudo-utility function with terms in the argument of the utility, we show that we can form a second algorithm, $\delta$-EigenGame, whose updates are the gradients of a proper utility function; remarkably, maximising this utility does indeed correspond to the GEP.

These changes now clearly articulate our work as a bridge between Hebbian and EigenGame approaches.

Interestingly, noting that optimizing the sum of the top-k utilities gives the top-k subspace of a GEP, and that for CCA our utility function consists entirely of covariances/inner products, we can apply our method to Deep CCA.

# 2) Improvement with respect to previous work

## GHA-GEP and $\delta$-EigenGame
While we recognise that our method shows a modest improvement over prior work in terms of accuracy and runtime for stochastic GEPs when all methods are fully tuned, we argue that hyperparameter tuning is also a component of the overall runtime for an empiricist. Since our model only requires tuning of the learning rate, this is a more powerful advantage over prior work than it may seem at first glance. Indeed, in the production of this paper we required around 3 times more compute to tune \gamma-EigenGame than our method owing to its 3 hyperparameters.

## Deep CCA
Uniquely among recent Hebbian and EigenGame approaches, we can extend our method to ‘Deep’ GEPs like DCCA. This is a direct result of our choice of geometry i.e. we do not constraint to the unit sphere. This opens up the possibility of using principled extensions of other classical algorithms to learn deep representations.

# 3) Motivations
We have significantly altered the communication of the paper's motivation. In particular we have motivated the paper specifically by GEPs associated with dimensionality reduction methods where it is natural in a streaming data setting to have sample approximations of the relevant GEP (but without loss of generality!).

DCCA has been held back in applications by needing very large batch sizes to obtain reasonable solutions. For memory intensive problems like imaging and biomedical data (where deep representations would otherwise be desirable), these large batch sizes are impractical. An effective DCCA would be useful in many multiview machine learning applications because CCA is a principled way of encouraging the consensus principle (NGuyen); maximizing the agreement among multiple representations of the data.

# 4) Conclusion
We would like to once again thank the reviewers for their helpful comments and for their time. We believe we have made substantial changes to the paper such that we better describe the motivation and proposed solutions. Given the significant improvement from the original submission, we would like to ask the reviewers to consider improving their scores for the revised paper.

While reviewers raised no concerns about reproducibility we would also like to add that we would be delighted to share the code for the paper.

---

### Decision · Program_Chairs · 2023-01-20

**Decision:**

Reject

**Justification For Why Not Higher Score:**


The proposed algorithm has been investigated in literature before in a more general setting.

**Justification For Why Not Lower Score:**

N/A

**Metareview: Summary, Strengths And Weaknesses:**


In this paper, the authors generalized the Hebbian rule through the corresponding Lagrangian of the Rayleigh Quotient form, which leads to a stochastic algorithm for generalized eigenvalue problem, including canonical correlation analysis and partial least square.

The major issue of the paper is the novelty. These generalized Hebbian rule has been proposed in kernel community [1], in which the finite-sample convergence has already been proved in a more general setting in RKHS. The proposed ignored this important literature. Despite the missing reference, the algorithm is only a small modification of Hebbian rule in [2].

Another issue is the clarity of the presentation. All the reviewers raised the confusion about the algorithm description and motivation.

In sum, the paper is not ready to be published yet in current version.

[1] Xie, Bo, Yingyu Liang, and Le Song. "Scale up nonlinear component analysis with doubly stochastic gradients." Advances in Neural Information Processing Systems 28 (2015).
[2] Terence D Sanger. Optimal unsupervised learning in a single-layer linear feedforward neural network. Neural networks, 2(6):459–473, 1989.